# OpenReview forum: "What is in Your Safe Data? Identifying Benign Data that Breaks Safety"
_colmweb.org/COLM/2024/Conference — COLM_

### Official Review · Reviewer_bd8E · 2024-05-08

**Rating:** 7
**Confidence:** 3
**Ethics Flag:** 1

**Summary:**

This paper takes a data-centric approach to explore why benign fine-tuning facilitates jailbreaks. To this end, representation matching and gradient matching are proposed to identify harmful samples in benign data. Finally, extensive experiments were conducted to prove the superiority of the proposed strategy, and it was found that lists and bullet points or math questions were more likely to lead to jailbreak.

**Questions To Authors:**

Refer to Reason To Reject.

**Reasons To Accept:**

1. The proposed method is simple and easy to understand, which makes it suitable for LLMs with high complexity.
2. Extensive experiments were conducted, and the experimental results proved that the proposed method seems to be effective.
3. Revealed that jailbreak data are often in the form of lists and bullet points, or math questions.

**Reasons To Reject:**

1. Evaluation metrics need a more detailed description.
2. Only using Random as the baseline, can other data selection strategies be considered, such as methods based on anomaly detection, or some strategies mentioned in related work.
3. Selected samples can cause greater harm than explicitly harmful samples and should be further discussed. Because the proposed method is based on similarity with explicitly harmful samples, theoretically, the performance is only close to them.
4. Smaller batch sizes are more harmful. Is there any further explanation for this phenomenon? Is this relevant to the migrability of adversarial attacks?
5. What is the Full Dataset of Table 1?

---

> ### Author Rebuttal · Authors · 2024-05-31
>
> We thank the reviewer for their careful review & for highlighting the clarity of our method, as well as extensive experiments. We provide responses to reviewer’s questions below and will update the paper accordingly.
>
> A1: Our evaluation is on a benchmark of 520 harmful questions from Zou et al., (2023). We adopt two methods for evaluation (see end of page 4 for more details): (1) Keyword Matching: this is a basic approach, matching whether there are keywords indicating refusal in the response, such as “I am sorry”, “I cannot”. This does not capture scenarios such as repeating the prompt without a meaningful harmful response. Therefore, we refine evaluation with (2) GPT Evaluation: the detailed prompt we use for the GPT scorer can be found in Appendix A.10. This metric gives detailed guidance for how to rate a response’s harmfulness from 1 to 5. We pay special attention to surface-level compliance such as repetition or vagueness, and assign them a low harmfulness score. We report both the average harmfulness score across the benchmark as well as GPT-ASR, which is the percentage of responses that receive a score of 5.
>
> A2: We ran additional experiments for BM25 selection, which has an average harmfulness score of 1.48 and average ASR of 10.72% after fine-tuning across 5 runs.
>
> A3: Our selection strategy is not a 1-to-1 matching of gradients. We are matching data as best as possible from a pool of benign data so that they result in potentially very similar models to harmful fine-tuning, but the top-ranked samples & gradients can be different from the original harmful data (while moving in the general direction of minimizing harmful task loss). This perturbation to the gradients means that their effect on the model is not upper-bounded/ restricted by the original dataset and might lead to a deeper jailbreak (since we evaluate on a hold-out set).
>
>
> A4: We observe that smaller batch sizes could be more harmful than larger ones, possibly because our selection mechanism compares individual data points’ gradient similarity for ranking. When using a smaller batch size for model updates, the impact of selected individual samples is more pronounced, as more data in a batch leads to a more smoothed averaged gradient. Future work might view this as a “slate” selection problem, taking into account batching effects in addition to datapoint similarity.
>
> A5: Full Dataset in Table 1 refers to using the entire Alpaca, or entire Dolly dataset to fine-tune the model.

---

> > ### Comment · Reviewer_bd8E · 2024-06-05
> >
> > I appreciate the author's response and I will maintain my positive view of the paper.

---

### Official Review · Reviewer_MHua · 2024-05-09

**Rating:** 8
**Confidence:** 4
**Ethics Flag:** 1

**Summary:**

This paper investigates a critical issue in the field of natural language processing (NLP), specifically focusing on the susceptibility of Large Language Models (LLMs) to jailbreaking. The authors explore why benign fine-tuning data can inadvertently degrade the safety of LLMs. The proposed methods of representation and gradient matching are interesting and useful for identifying potentially harmful benign data. More empirical analyses consolidate the findings and provide valuable insights.

**Reasons To Accept:**

* The paper targets an important problem in LLM safety, e.g., how benign data leads to jailbreaking after fine-tuning.
* The technical contributions are reasonable. The authors introduce a bi-directional anchoring method to identify benign data that can degrade model safety.
* Extensive experiments provide valuable insights.

**Reasons To Reject:**

* While the paper claims to identify subsets of benign data that can compromise safety, the analysis may not be exhaustive. The authors acknowledge that other metrics could potentially improve on their approaches, indicating a space for further research.
* The experiments are primarily conducted on two LLMs from the LLAMA family. The generalizability of the findings to other models and datasets is not fully established.
* The paper could benefit from a deeper theoretical analysis of why certain benign data subsets lead to jailbreaking. A theoretical framework would strengthen the claims made by the empirical results.

Despite the above, I am sure that the paper has made enough contributions to reach the publication bar. It can inspire future work to address these weaknesses.

---

> ### Author Rebuttal · Authors · 2024-05-31
>
> We appreciate the reviewer’s detailed comments and thank the reviewer for highlighting the insights and contribution of our work. We will update our work based on reviewer feedback. We add additional experiments on newer models; provide some intuition to help explain the empirical results.
>
> Experiments: Our main experiments were on Llama 2 mainly due to the limited open-source strongly-aligned models available at the time. We find that previously-selected Alpaca data also transfers to Gemma 7b Instruct and Llama 3 8b Chat models. From random to representation selection: Llama3 (25.9%->57%); Gemma (34.2%->49.3%). From random to gradient selection: Llama3 (->43.31%); Gemma (->48.8%). We will include more details for these in the final paper. This shows that our selected data/methods transfer to other models. We also note that we already include another dataset: GSM8k in Section 4.2.
>
> Understanding:  We will add more discussion for why these patterns are more likely to lead to jailbreaks. Briefly, if cosine similarity of the harmful data & selected data is positive, they share a descent direction, eg, Farajtabar et al (2020). Initial examination suggests that lists/math data have closer positive similarity to harmful gradients (Lists: 0.05, GSM8k: 0.01) compared to random selections, which have negative similarity on average (-0.0395 +/- 0.0032). Selected data may share similar latent qualities (eg, responsiveness to user requests) with the harmful data. Recent work, for example, suggests that early tokens play a larger role in alignment/finetuning (Zhang and Wu, 2024; Lin et al., 2023). Matching harmful data may move the model toward an optimum with a prefix other than a refusal (eg, “I’m sorry..” -> “1. The first step is..”), partially unlearning previous alignment.
>
>
> References:
> M. Farajtabar et al., “Orthogonal gradient descent for continual learning,” in AISTATS, 2020.
>
> Xiao Zhang and Ji Wu. Dissecting learning and forgetting in language model finetuning. In The Twelfth International Conference on Learning Representations, 2024.
>
> Bill Yuchen Lin, Abhilasha Ravichander, Ximing Lu, Nouha Dziri, Melanie Sclar, Khyathi Chandu, Chandra Bhagavatula, and Yejin Choi. The unlocking spell on base llms: Rethinking alignment via in-context learning. arXiv preprint arXiv:2312.01552, 2023.

---

> > ### Comment · Reviewer_MHua · 2024-06-03
> >
> > Thanks for the authors' response. I will maintain the rating.

---

### Official Review · Reviewer_Zt4p · 2024-05-09

**Rating:** 7
**Confidence:** 3
**Ethics Flag:** 1

**Summary:**

Summary: this paper investigates data that at a first glance seems to be harmless and of good quality, namely benign data, but in reality it poses a threat to the safety of LLMs by acting as a jailbreaker.

Empiricism, Data, and Evaluation: the experiments seem to be well designed and carried out; the data used seems to be appropriate, as well as the evaluation metrics.

Technological Impact: the code will be available which would be beneficial for people testing the safety of their own datasets.

Ambition, Vision, Forward-outlook: the vision of this paper is somewhat shortsighted since this is not a new problem, and the methods used are basically variations of previous methods, though the paper contributes with a well-focused and narrow research question about what type of benign data elicits harmful behavior from LLMs.

Understanding Depth, Principled Approach: the paper presents some analysis about the type of data that breaks LLMs; however, there is no explanation (neither understanding) of why this data that seems harmless is actually a jailbreaker.

Clarity, Honesty, and Trust: the paper is well-written and is clear in most parts, and it seems to be trustable due to their concise report of results and reproducibility parameters.

**Questions To Authors:**

1) can you elaborate on the rationale behind the Feature Definition for the gradient-based similarity approach?

2) can you elaborate on the results from Table 1 about the Pure-bad dataset? Are these the results after fine-tuning on the whole gold harmful data?

3) do you have any explanation or logic behind why the data in the form of lists and math equations break the safety of Llama 2?

**Reasons To Accept:**

This paper proposes a concise research question and the corresponding experiments to answer it. The baselines proposed are acceptable; the research question is answered by showing how particular instances (prompts soliciting lists or mathematical derivations) are the main causes for breaking the safety of LLMs. Moreover, the methods proposed to find these instances seem adequate: 1) similarity of candidate instances with respect to gold harmful data in terms of the parameters of the LLM (the last layer) after processing the instance, and 2) similarity of candidate instances to 2 gradient vectors (the gradient vector of gold harmful instances and the vector of gold safe instances). Finally, the code will be released which could be very useful for the community.

**Reasons To Reject:**

First, the scope is narrow since experiments are done only for Llama 2 (7B and 13B); moreover, only one harmful dataset is used and only 2 fine-tuning datasets are assessed (Alpaca and Dolly). This setup, even though well-thought, makes results quite restricted; thus, the paper looks more like a specific case study, which should be presented as such to better contextualize the reader, and probably would better suit a short paper. Furthermore, and in my opinion the most important point, there is no explanation of why data in the form of lists and mathematical equations elicit the jailbreaking phenomenon in Llama 2; this is a very interesting discovery but no explanation, no rationale, and therefore no attempt for understanding is carried out.

---

> ### Author Rebuttal · Authors · 2024-05-31
>
> We thank the reviewer for their thoughtful & positive comments. To address reviewer feedback, we provide additional experiments & discussion below. We will include these in the final paper.
>
> We will add caveats to our final paper as suggested by the reviewer. Our main experiments were on Llama 2 mainly due to the limited open-source strongly-aligned models available at the time. We find that previously-selected Alpaca data also transfers to Gemma 7b Instruct and Llama 3 8b Chat models. From random to representation selection: Llama3 (25.9%->57%); Gemma (34.2%->49.3%). From random to gradient selection: Llama3 (->43.31%); Gemma (->48.8%). We will include more details for these in the final paper. This shows that our selected data/methods transfer to other models. We also note that we already include another dataset: GSM8k (Section 4.2).
>
> Understanding:We will add more discussion for why these patterns are more likely to lead to jailbreaks. Briefly, if cosine similarity of the harmful data & selected data is positive, they share a descent direction, eg, Farajtabar et al (2020). Initial examination suggests that lists/math data have closer positive similarity to harmful gradients (Lists: .05, GSM8k: .01) compared to random selections, which have negative similarity on average (-.0395 +/- .0032). Selected data may share similar latent qualities (eg, responsiveness to user requests) with the harmful data. Recent work, for example, suggests that early tokens play a larger role in alignment/finetuning (Zhang and Wu, 2024; Lin et al., 2023). Matching harmful data may move the model toward an optimum with a prefix other than a refusal (eg, “I’m sorry..” -> “1. The first step is..”), partially unlearning previous alignment.
>
> A1: Rationale: If similarity is large enough, these benign examples update the model similarly to finetuning on harmful data (reducing loss on these harmful examples’ completions). The resulting loss decrease on the harmful data should increase the likelihood of generating harmful data/reduce likelihood of a safety response.
> A2: Yes, these results are after finetuning on the entire gold harmful data, which has 100 harmful questions paired with harmful responses. This sample size of 100 is consistent across the other fine-tuning setup where we randomly select 100 examples, use representation-matching method to select 100 examples, or use gradient-matching method to select 100 examples.
> A3: See the second part of our “Understanding” section above.

---

> ### Comment · Reviewer_Zt4p · 2024-06-04
>
> Thanks to the authors for their reply, I have read it and my score remains the same.

---

### Official Review · Reviewer_wt8f · 2024-05-10

**Rating:** 5
**Confidence:** 4
**Ethics Flag:** 1

**Summary:**

This paper proposes a framework for selecting abnormal data points that seems to be benign but can actually degrade the performance. The data points are ranked based on two different types of scores: (1) the inter product of representation vectors and (2) the inter product of the first-order gradient vectors. The authors demonstrate that these scores can be used to identify the seemingly benign but harmful data points on a couple of datasets.

---

I noted the authors' response and would like to raise my rating by 1, as the authors promise that they will provide more theoretical discussion.

**Questions To Authors:**

For the gradient matching, only the first-order gradients are considered. However, if the loss converges, should not the first-order gradients be very close to zero? Then why not consider the second-order gradient terms?

**Reasons To Accept:**

The idea of identifying the data points that may cause negative impacts on the model performance is interesting. The idea of utilizing influence function seems to be a novel contribution, at least within the field of fine-tuning language models.

**Reasons To Reject:**

It is not clear why the similarity of representation features or gradients can be used to identify the harmful data points. I would expect some theoretical justifications.

---

> ### Author Rebuttal · Authors · 2024-05-31
>
> We thank the reviewer for their thoughtful comments and for noting the novelty in our approach. We address the reviewer's concerns in two parts:
>
> (1) Why do these methods work?
> We will add more discussion in our final paper for why these patterns are more likely to lead to jailbreaks. Briefly, if cosine similarity of the harmful data & selected data is positive, they share a descent direction, eg, Farajtabar et al (2020). Initial examination suggests that lists/math data have closer positive similarity to harmful gradients (Lists: 0.05, GSM8k: 0.01) compared to random selections, which have negative similarity on average (-0.0395 +/- 0.0032). Selected data may share similar latent qualities (eg, responsiveness to user requests) with the harmful data. Recent work, for example, suggests that early tokens play a larger role in alignment/finetuning (Zhang and Wu, 2024; Lin et al., 2023). Matching harmful data may move the model toward an optimum with a prefix other than a refusal (eg, “I’m sorry..” -> “1. The first step is..”), partially unlearning previous alignment. Essentially, if selected benign data gradients are similar enough to harmful gradients, they are more likely to move in the same direction as a jailbreak (which is known to occur on the harmful data).
>
> (2) Why first-order gradients?
> In reality, the language modeling loss seldom fully converges; instead, it continues to decrease as more data or additional training epochs are introduced to the model. Therefore, the gradient does not achieve zero, especially on fine-tuning data points (Alpaca or Dolly) it has not seen before. Furthermore, second-order gradient terms are problematic for two reasons: (1) They are prohibitively expensive in the context of large language models with billions of parameters, and (2) they have not been demonstrated to be beneficial for data selection in language models to justify the high cost. However, we appreciate that this is an interesting future direction, and have already noted this in our future work section.
>
> References:
> M. Farajtabar et al., “Orthogonal gradient descent for continual learning,” in AISTATS, 2020.
>
> Xiao Zhang and Ji Wu. "Dissecting learning and forgetting in language model finetuning". In The Twelfth International Conference on Learning Representations, 2024.
>
> Bill Yuchen Lin et al., "The unlocking spell on base llms: Rethinking alignment via in-context learning". arXiv preprint arXiv:2312.01552, 2023.

---

### Decision · Program_Chairs · 2024-07-10

**Decision:**

Accept

**Comment:**

Authors investigate why finetuning on benign data can degrade the safety of LLMs and proposes ways to mitigate the issues, by successfully identifying potentially harmful benign data. Extensive experiments show the effectiveness of the approach. All reviewers have lauded the clarity, extensive experiments and the timely studied problem. A good paper.